# In situ and transcriptomic identification of microglia in synapse-rich regions of the developing zebrafish brain

Nicholas J. Silva [1,4], Leah C. Dorman [1,2,4], Ilia D. Vainchtein[1], Nadine C. Horneck [1] & Anna V. Molofsky [1,3✉]

Microglia are brain resident macrophages that play vital roles in central nervous system (CNS) development, homeostasis, and pathology. Microglia both remodel synapses and engulf apoptotic cell corpses during development, but whether unique molecular programs regulate these distinct phagocytic functions is unknown. Here we identify a molecularly distinct microglial subset in the synapse rich regions of the zebrafish (*Danio rerio*) brain. We found that ramified microglia increased in synaptic regions of the midbrain and hindbrain between 7 and 28 days post fertilization. In contrast, microglia in the optic tectum were ameboid and clustered around neurogenic zones. Using single-cell mRNA sequencing combined with metadata from regional bulk sequencing, we identified synaptic-region associated microglia (SAMs) that were highly enriched in the hindbrain and expressed multiple candidate synapse modulating genes, including genes in the complement pathway. In contrast, neurogenic associated microglia (NAMs) were enriched in the optic tectum, had active cathepsin activity, and preferentially engulfed neuronal corpses. These data reveal that molecularly distinct phagocytic programs mediate synaptic remodeling and cell engulfment, and establish the zebrafish hindbrain as a model for investigating microglial-synapse interactions.

[1] Department of Psychiatry and Behavioral Sciences/Weill Institute for Neurosciences, University of California, San Francisco, San Francisco, CA, USA.
[2] Neuroscience Graduate Program, University of California, San Francisco, San Francisco, CA, USA. [3] Kavli Institute for Fundamental Neuroscience, University of California, San Francisco, San Francisco, CA, USA. [4] These authors contributed equally: Nicholas J. Silva, Leah C. Dorman. ✉email: anna.molofsky@ucsf.edu

Microglia are the dominant immune cells in the central nervous system (CNS). They perform critical functions during brain development and disease including engulfing synapses, promoting synapse formation, and clearing apoptotic neurons[1,2]. However, it is not clear whether these diverse functions are mediated by molecularly distinct subsets of microglia. Single-cell sequencing of mouse microglia reveals transcriptional heterogeneity predominantly during development[3,4], whereas functional studies reveal region-specific microglial populations that persist into adulthood[5–7]. These data suggest that transcriptional profiling in rodents has not yet been able to resolve known functional heterogeneity in microglia. Furthermore, linking these transcriptionally identified subpopulations to functional subsets in situ remains challenging. Thus, despite abundant evidence that microglia both modulate synapses and engulf cell corpses during development, the molecular regulation of these different functions is not well understood.

The zebrafish (*Danio rerio*) is an increasingly utilized vertebrate model for developmental neuroscience[8] that has not yet been used to study microglial–synapse interactions. Zebrafish microglia are ontogenetically similar to mammalian microglia and express most canonical microglia genes identified in mammals[9–12]. The neuroimmune interface is also similar between species, including a diversity of glial cell types[13–15] and an immune system homologous to mammals[16–18], including meningeal lymphatics[19] and a blood-brain barrier that matures between 8 and 10 days post fertilization (dpf)[20]. There are also many similarities in nervous system development, with the exception that robust neurogenesis persists throughout life in the zebrafish brain. This leads to ongoing turnover of neuronal corpses in neurogenic regions, particularly in the midline optic tectum (OT). Much of our functional understanding of fish microglia comes from elegant work in the OT identifying molecular mechanisms that drive phagocytosis of neuronal corpses[21–23]. However, microglia in the OT are functionally and ontogenetically distinct from those found in other CNS regions[24–27]. In addition, a subset of microglia in the spinal cord white matter engulfs myelin sheaths[28] and is linked to leukodystrophy[29]. These functionally distinct microglia subsets coexist in the developing zebrafish nervous system and are to some extent segregated by the CNS region.

Interestingly, despite tremendous interest in defining the role of microglia in synaptic remodeling in mammals, it is not known whether there are unique molecular programs that subserve this functional specialization. Furthermore, although the zebrafish is an ideal model system in which to study microglial-synaptic interactions, whether microglia are present in synaptic regions of the fish CNS is not well defined. Recent studies suggest that transcriptional and functional heterogeneity of microglia in the adult zebrafish brain is in part linked to their developmental origins, whereby ameboid OT microglia derived from the rostral blood island and aortic gonad mesonephros (AGM)[30–32] have enhanced capacity to phagocytose bacteria in vitro, compared with more ramified microglia that appear to be exclusively AGM derived[32]. Like in mammals, it is difficult to precisely define to what extent ontogeny defines function. However, these studies suggest that zebrafish microglia are regionally and molecularly distinct in a manner that impacts function, raising the question of whether synapse-associated functions could be transcriptionally defined.

Here, we identify a distinct subpopulation of synaptic region-associated microglia (SAMs) in the juvenile fish using in situ characterization as well as single-cell and bulk RNA sequencing. SAMs were ramified and expanded in the midbrain and hindbrain after 7 dpf. These microglia engulf neuronal synaptic proteins as in mammals and were defined by the expression of complement genes (*c1qa, c1qc*) and other candidate pathways. In contrast, microglia in the OT clustered near neurogenic regions, were rich in lysosomal gene expression (*ctsla, ctsba*), and their phagocytic capacity correlated with functional cathepsin activity. These data define a molecular profile associated with phagocytosis of synaptic proteins and suggest a model system in which to study microglial–synapse interactions.

## Results

**Microglia in synaptic regions expand developmentally in the zebrafish hindbrain and are distinct from neurogenic-associated microglia.** To characterize microglia in both synaptic and neurogenic regions of the fish brain, we performed immunohistochemistry (IHC) using the myeloid reporter line *Tg(mpeg1.1:GFP-CAAX)*[21]. We used the presynaptic vesicle marker SV2 to demarcate synapse-rich regions and bromodeoxyuridine (BrdU) labeling to identify neurogenic zones. We observed microglia in synapse-rich regions of the midbrain and hindbrain at the earliest time-point we examined, 7 days post fertilization (dpf), and found that they increased in number over the late larval to the juvenile period (14–28 dpf; Fig. 1a, b). By 28 dpf most microglia in the midbrain and hindbrain colocalized with the presynaptic marker SV2 (65% and 80%, respectively; Fig. 1c, d). In contrast, >60% of microglia in the OT clustered near the BrdU+ neurogenic zone (Fig. 1e, f), consistent with their known roles in eliminating apoptotic cell corpses[25,27]. Immunostaining with the commonly used microglia antibody 4C4 was consistent with these findings and indicated that the majority (70–90%) of *mpeg1.1*-GFP[+] cells in the brain are microglia, with a modestly lower proportion in the midbrain and hindbrain relative to OT (Fig. S1a, b). Of note, although 4C4 is not detected in border-associated macrophages, it also fails to label some ramified *mpeg1.1*-GFP[+] parenchymal cells that are putative microglia, suggesting that it is specific to microglia (vs. macrophages), but not entirely sensitive (Fig. S1c–e). We next characterized microglial morphology in the hindbrain and OT. We found that microglia (*mpeg1.1*-GFP[+] 4C4[+]) in the hindbrain were ramified and more closely resembled microglia in the postnatal rodent brain, whereas OT microglia were on average significantly more ameboid, as quantified by increased sphericity (Fig. 1g, h). Sholl analysis supported the finding that HB microglia is significantly more ramified than OT microglia (Fig. 1i). In summary, we identified ramified microglia enriched in synapse-rich regions of the hindbrain and midbrain that were distinct from ameboid microglia seen around neurogenic regions of the OT.

**Molecularly distinct subsets of microglia identified at single-cell resolution during brain development.** To determine whether microglia in the zebrafish brain are molecularly heterogeneous, we performed single-cell RNA sequencing. We flow-sorted hematopoietic cells from juveniles at 28 dpf using a *Tg(cd45:DsRed)* reporter[33] crossed with the myeloid-specific *Tg(mpeg1.1:EGFP)* reporter to ensure that all potential subsets of immune cells were captured (Fig. 2a, S2a). This juvenile time-point of 28 dpf encompasses both developmental waves of embryonic and adult microglia in the zebrafish brain[30,31]. Flow analysis indicated that 90% of *cd45*-DsRed[+] cells were *mpeg1.1*-EGFP[+], and that the *cd45*-DsRed[+] population captured all *mpeg1.1*-EGFP[+] cells. Unbiased clustering following single-cell RNA sequencing of 6666 *cd45*-DsRed[+] cells revealed 15 distinct clusters of hematopoietic origin, including seven non-myeloid clusters (*mpeg1.1* negative; Fig. 2b, S2b, c; Supplementary Data S1). These included clusters that expressed markers for T-cells (*cd4-1, lck*, and *ccr9a*), natural killer cells (*eomesa*), and

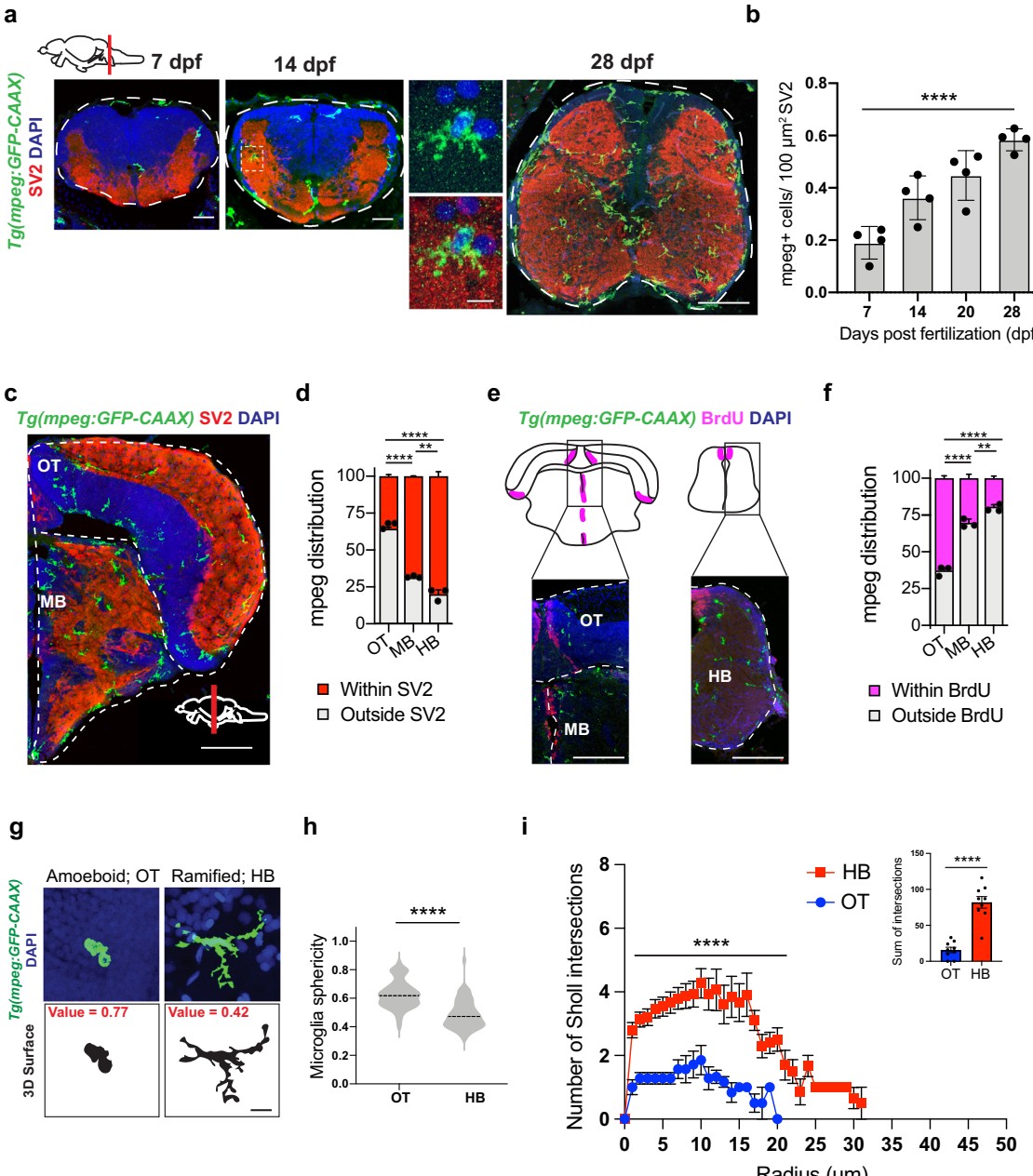

**Fig. 1 Phenotypically distinct populations of microglia in synaptic and neurogenic regions of zebrafish brain. a** Representative images of microglia (*mpeg1.1-GFP+*) and synapses (SV2 antibody stain) in developing zebrafish hindbrain. Inset: representative synapse-embedded microglia at 14 dpf. Scales: 7–14 dpf, 20 μm; 28 dpf, 50 μm; inset, 5 μm. All images are representative of the $n = 4$ replicates. **b** Quantification of *mpeg1.1-GFP+* cells per 100 μm$^2$ of SV2+ synaptic area at 7, 14, and 28 days post fertilization. Dots represent four individual fish, data are mean ± SD. One-way ANOVA *$p < 0.0001$. **c–d** Representative images and quantification of the proportion of *mpeg1.1-GFP+* cells in synaptic (SV2+; red) vs. cellular (DAPI+/SV2−; gray) areas. Distribution quantified within each brain region as outlined (dotted lines): midbrain (MB), optic tectum (OT), and hindbrain (HB, see panel 1 A). Mean ±SEM from $n = 3$ fish. Two-way ANOVA with Tukey's post hoc comparison; ****$p < 0.0001$, **$p < 0.0029$. Scale: 50 μm. **e–f** Representative images and quantification of the proportion of *mpeg1.1-GFP+* cells within 20 μm of BrdU+ neurogenic regions (purple) vs. outside neurogenic regions (gray). Mean ±SEM from $n = 3$ fish. Two-way ANOVA with Tukey's post hoc comparison; ****$p < 0.0001$, **$p < 0.0062$. Scale: 50 μm. **g** Representative images of *mpeg1.1-GFP+* microglia and thresholded maximal projections in OT and HB at 28 dpf. Value = sphericity; scale 0–1, 1 = most spherical/ameboid. 5 μm. Images are representative of respective values indicated in red. **h** Quantification of microglial sphericity from images thresholded as in **g**. $n = 50$ microglia per region from $n = 4$ fish. The dotted line indicates the median. two-tailed unpaired *t* test; ****$p < 0.0001$ **I** Sholl analysis quantifying the number of intersections (*y* axis) measured at 1 μm increments from the soma (*x* axis) in microglia from optic tectum (OT) and hindbrain (HB). Total of $n = 9$ microglia per region from $n = 3$ fish. Mean ±SEM. Two-way ANOVA with Sidak's multiple comparisons; ****$p < 0.0001$. Inset: two-tailed unpaired *t* test; ****$p < 0.0001$. See also Fig. S1.

innate lymphocyte-like cells *(il13, gata3)*[34–37]; Fig. S2c). Thus, multiple immune cell subsets, predominately myeloid in origin, are present in the zebrafish brain, although it is possible that some of these may be circulating rather than tissue-resident.

To focus on myeloid cells, which are the dominant immune cell subset, we reclustered 3539 *mpeg1.1+* cells. After quality control and filtering, the myeloid subset yielded six distinct clusters (Fig. 2c; Fig. S2d–g; Supplementary Data S2) at a clustering

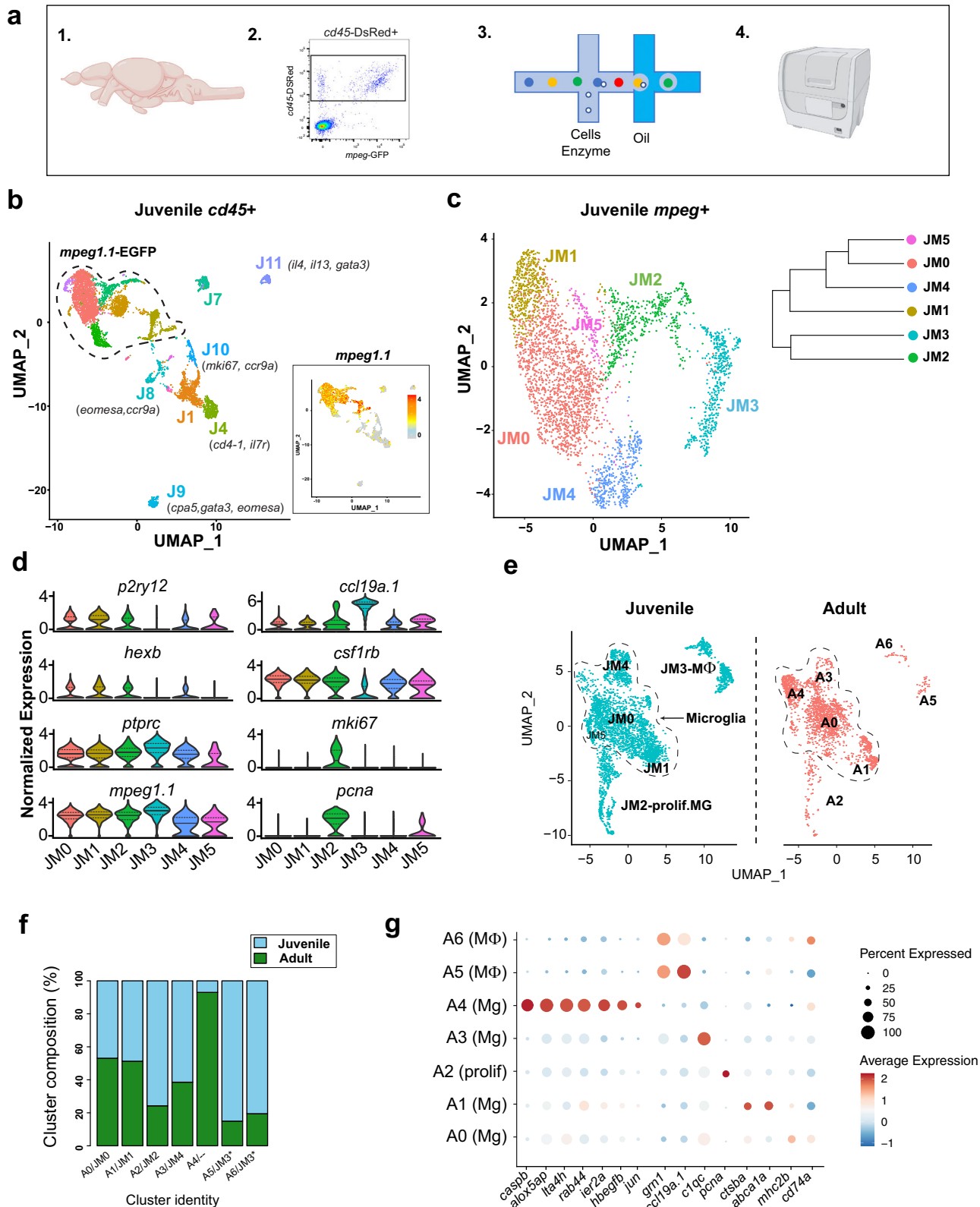

resolution of 0.3, which best represented the observed patterns of gene expression (Fig. S2h, i). We identified cluster JM3 as the macrophage subset based on the absence of microglial-specific markers (*p2ry12, csf1ra, hexb*, and *slc7a7*), the presence of *ccl19a.1 (a.k.*a. macrophage inflammatory protein), *flt3*, and *grn1* (Fig. 2d, Fig. S3a), and higher levels of *mpeg1.1* (Fig. S2g). However, we could not definitively identify whether the JM3

macrophage cluster represented CNS-resident 'border associated' macrophages[38] vs. circulating macrophages, as the zebrafish homologs of mammalian BAM genes (e.g., *cd63, mrc1b, apoc1, apoc4*, and *lyve1b*)[39] were either not detected or did not segregate as expected between microglial and macrophage subsets (Fig. S3a). Of the remaining microglial clusters, cluster JM2 marked a proliferating subset (*pcna* and *mki67*; Fig. 2d), cluster JM5

**Fig. 2 Functionally distinct subsets of microglia identified at single-cell resolution during brain development and in adulthood. a** Schematic diagram of the whole-brain scRNA-sequencing pipeline (items 1 and 4 adapted from BioRender.com (2020). Retrieved from https://app.biorender.com/biorender-templates. **b** Unsupervised clustering of 6666 juvenile *ptprc (cd45)*+ cells, colored by cluster. Inset: feature plot of *mpeg1.1* expression. $n = 10$ fish in two independent replicates. Color-coding for inset: normalized *mpeg1.1* expression; low expression, gray; high expression, orange. **c** Subclustering of 3539 *mpeg1.1*+ cells in **c**, colored by cluster. Inset: cluster dendrogram. **d** Violin plots for select marker genes across juvenile clusters from **c** (JM1-5), including microglia (*p2ry12, hexb,* and *csf1rb*), pan-myeloid (*mpeg1.1*), pan-hematopoietic (*ptprc/cd45*), macrophage (*ccl19a.1*) and proliferative (*mki67* and *pcna*). Colors correspond with clusters shown in Fig. 2c. (Solid black line = median; dotted lines = 1st and 3rd quartiles; any lines not visible fall at 0). **e** UMAP plots showing co-clustering analysis of juvenile cells in **c** (blue) and 2080 adult (red) myeloid (*mpeg1.1*+) cells. Data sets were merged then integrated with the Harmony R package before clustering. Conserved macrophage and proliferative clusters indicated. The dotted line outlines the remaining "homeostatic" microglial clusters. **f** Percent of each cluster shown in **e** consisting of Juvenile (blue) vs. Adult (green) derived cells. Normalized within groups to account for overall cell number difference between ages. Asterisks indicate a shared juvenile cluster. **g** Dot plot highlighting select up- and downregulated genes within the combined juvenile and adult clustering in **e**. Size represents the percent of cells expressing each gene while the color represents normalized and scaled gene expression compared with all clusters; decreased expression: blue, expression unchanged: white, increased expression: red (MAST DE algorithm in Seurat, $p < 0.01$). See also Fig. S2 and Fig. S3; Supplementary Data S1–S3.

contained too few cells to be definitively assigned, and the remaining three clusters (JM1, JM0, and JM4) had a clear microglial identity but unique gene expression profiles which we further analyze below.

Next, we examined whether these subsets persist into adulthood by co-clustering the juvenile *mpeg1.1*+ population with adult (1-year-old) brain *mpeg1.1*+ cells that were sorted and sequenced in parallel. The datasets were integrated using the Harmony R package[40] to compensate for sources of technical variation. Unbiased clustering and differential expression analysis revealed six subsets that were conserved between juveniles and adults (Fig. 2e; Supplementary Data S3) as well as a distinct adult-enriched microglial cluster (A4; Fig. 2f). Notably, adult cluster A3 mapped to juvenile cluster JM4 (conserved genes included *cebpb, c1qa, c1qb,* and *c1qc*), whereas adult cluster A1 mapped to juvenile cluster JM1 (*apoeb* and *ctsba*). A proliferative cluster was still present (A2; *pcna, tubb2b,* and *mki67*), as well as two distinct macrophage clusters (A5 and A6) corresponding to juvenile cluster JM3 (*ccl19a, siglec15l,* and *cmklr1*). Interestingly, the adult-enriched cluster A4 was distinctly enriched in inflammasome genes, suggesting a microglial subset poised for inflammasome activation (*caspb, fads2,* and *alox5ap*; Fig. 2g). Taken together, these data reveal myeloid cell heterogeneity in the juvenile zebrafish brain that persists into adulthood.

**Region-specific transcriptional signatures of juvenile zebrafish myeloid cells.** Previous zebrafish transcriptomes using the *Tg(mpeg1.1:EGFP)* transgenic line and bulk-sequencing approaches have suggested regional transcriptional heterogeneity within the zebrafish brain[32]. To better compare our single-cell data with the existing literature[10,11], we isolated *mpeg1.1*-EGFP+ myeloid cells from the OT, midbrain, and hindbrain at 28 dpf by flow cytometry and performed bulk RNA sequencing (Fig. 3a, Fig. S4a, Supplementary Data S4). The resulting transcriptome was highly enriched for microglial-specific genes (*csf1rb, c1qa,* and *p2ry12*), as well as some macrophage markers (*siglec15l, spock3*). This was a pure myeloid population with no detectable evidence of neuronal, astrocyte, or oligodendrocyte genes (Fig. S4b). Principal component analysis revealed that 60% of gene variance was driven by brain region (PC1; Fig. 3b), and differential expression analysis of these subsets ($p < 0.05$) identified region-specific gene expression signatures (Fig. 3c). In particular, complement and antigen presentation genes were enriched in the hindbrain (*c1qa, c1qc, cd74a,* and *mhc2dab*), whereas lysosome-associated genes were highly enriched in the OT (*apoeb, ctsba, ctsc,* and *ctsla*). The midbrain was intermediate and did not clearly segregate with either phenotype.

We confirmed region-specific expression for two candidate genes from this profile via in situ hybridization (ISH). The

hindbrain-enriched gene *cd74a* was expressed in over 90% of hindbrain microglia (*mpeg1.1*-GFP+ 4C4+) and 30% of OT microglia, with intermediate expression in the midbrain (Fig. 3d, e). In contrast, the OT-enriched gene *ctsba* was highly enriched in neurogenic areas and highest in the OT compared with the midbrain and hindbrain (72%, 33%, 19%, respectively; Fig. 3f, g). These data reveal brain-region-specific transcriptional signatures that can be identified in situ. However, they also suggest that these differences may be partly driven by varying proportions of phenotypically distinct microglia as well as resident macrophages that are better discriminated at the single-cell level of resolution.

**Region-enriched functional microglial subsets in zebrafish hindbrain and OT.** We next used the region-specific transcriptional signatures to map the single-cell data, with the goal of localizing functional subsets in situ (Fig. 4a). To do this, we calculated an "eigengene" composed of the top differentially expressed genes in the *mpeg1.1*:EGFP+ population from each region. Overlay of these region-defining eigengenes onto our UMAP plot revealed enrichment of the OT signature in the juvenile cluster JM1, whereas the hindbrain eigengene was enriched in JM4 and the macrophage cluster JM3 (Fig. 4b, c). The midbrain regional signature was indeterminate and aligned only with the macrophage cluster, consistent with our finding of substantially more macrophages in this region (Fig. S1b). Cluster JM0 was not enriched in any regional signature, suggesting that it may represent either a common microglial population or a subset not represented in our bulk-sequencing. The OT-enriched cluster JM1 and the hindbrain-enriched cluster JM4 were distinct in both bulk and single-cell sequencing and did not contain contaminating macrophages. We therefore directly compared the transcriptomic profile of these subsets (Fig. 4d, e, Supplementary Data S5).

Genes that defined the hindbrain-enriched cluster (JM4) included *c1qa*, the initiating protein in the complement cascade and a known regulator of synaptic engulfment expressed in rodent microglia[41,42]. Another cluster-defining gene was *cebpb*, a transcription factor that promotes microglial homeostasis in neurodegenerative disease[43]. Other upregulated genes included the hindbrain region-defining gene *cd74b*, which encodes the invariant chain of major histocompatibility complex class II, and *grn1*, which encodes progranulin and interacts with C1q to regulate synaptic pruning[44]. We also identified several putative functional genes not previously studied such as the lysozyme gene *lygl1*. Gene ontology analysis suggested that the preferentially expressed genes in cluster JM4 are involved in protein production and metabolism, including translational elongation, negative regulation of proteolysis, and protein folding (Fig. 4d, e).

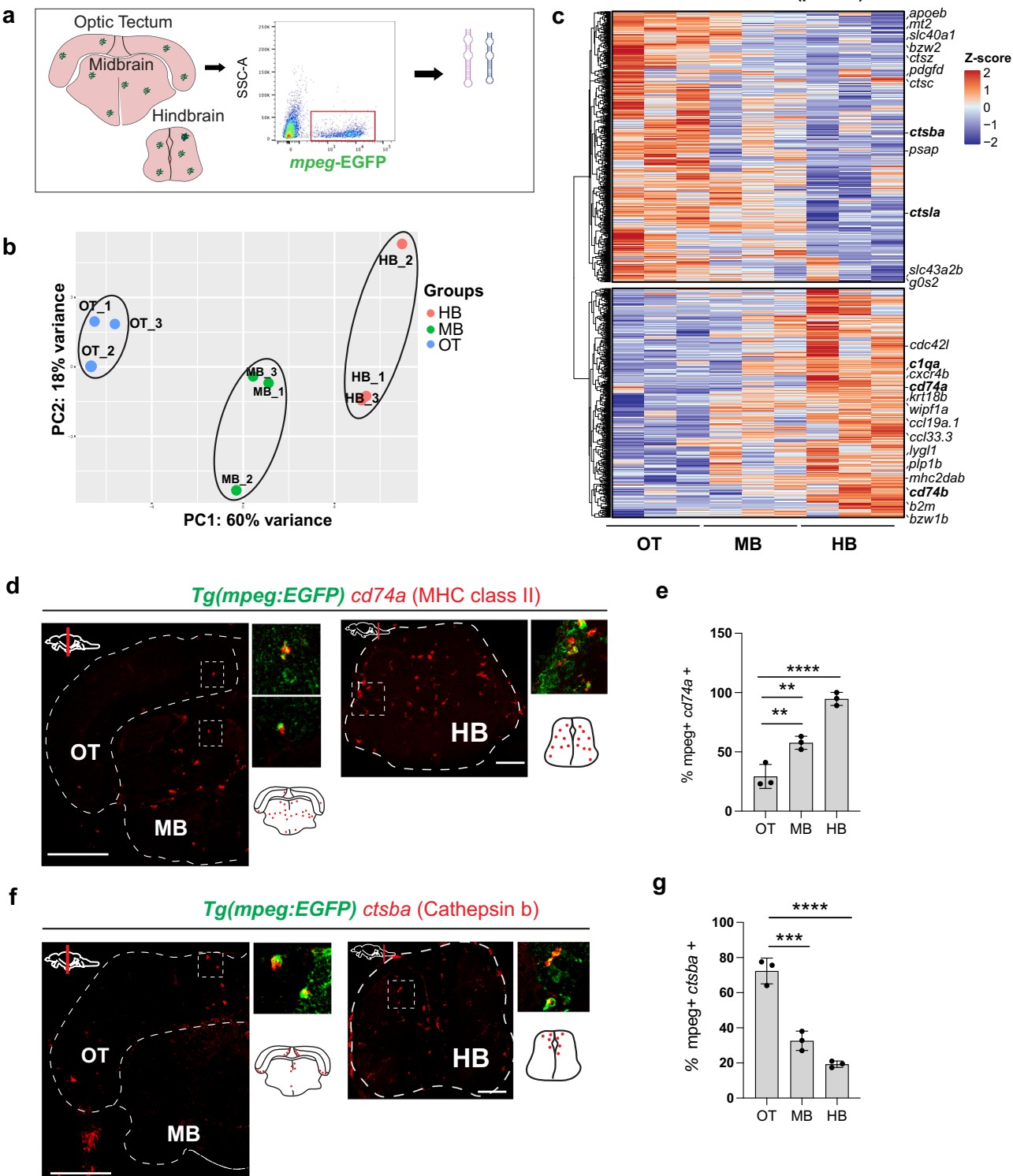

**Fig. 3 Region-specific transcriptional signatures of juvenile zebrafish myeloid cells. a** Schematic of region-specific bulk-sequencing approach using *Tg(mpeg1.1:EGFP)*. **b** Principal component analysis (PCA) plot of top 500 most variable genes for OT, MB, HB regions (dots = independent replicates from 10 pooled fish/replicate). **c** Heatmap of differentially expressed genes (DEGs) from *mpeg1.1*-EGFP positive cells across brain regions (adjusted *p* value < 0.05). Selected top differentially expressed genes from each cluster highlighted. Color-coding, decreased expression; blue, no expression; white, high expression; red. **d–e** Representative images and quantification of *cd74a* expression across brain regions by in situ hybridization (ISH), colocalized with *mpeg1.1*-GFP+. Quantification shows a percentage of *cd74+ mpeg1.1*-GFP+ microglia within each brain region. Dots represent three individual fish, data are mean ± SD. One-way ANOVA with Tukey's post hoc test. **p < 0.0079, ****p < 0.0001, **p < 0.0021. Scale: 50 μm. All images are representative of the n = 3 replicates. **f–g** Representative images and quantification of *ctsba* expression by ISH colocalized with *mpeg1.1*-GFP. Quantification shows percentage *ctsba+ mpeg1.1*-GFP+ microglia within each brain region. Dots represent three individual fish, data are mean ± SD. One-way ANOVA with Tukey's post hoc test. ***p < 0.0003, ****p < 0.0001. Scale: 50 μm. All images are representative of the n = 3 replicates. See also Fig. S4, Supplementary Data S4.

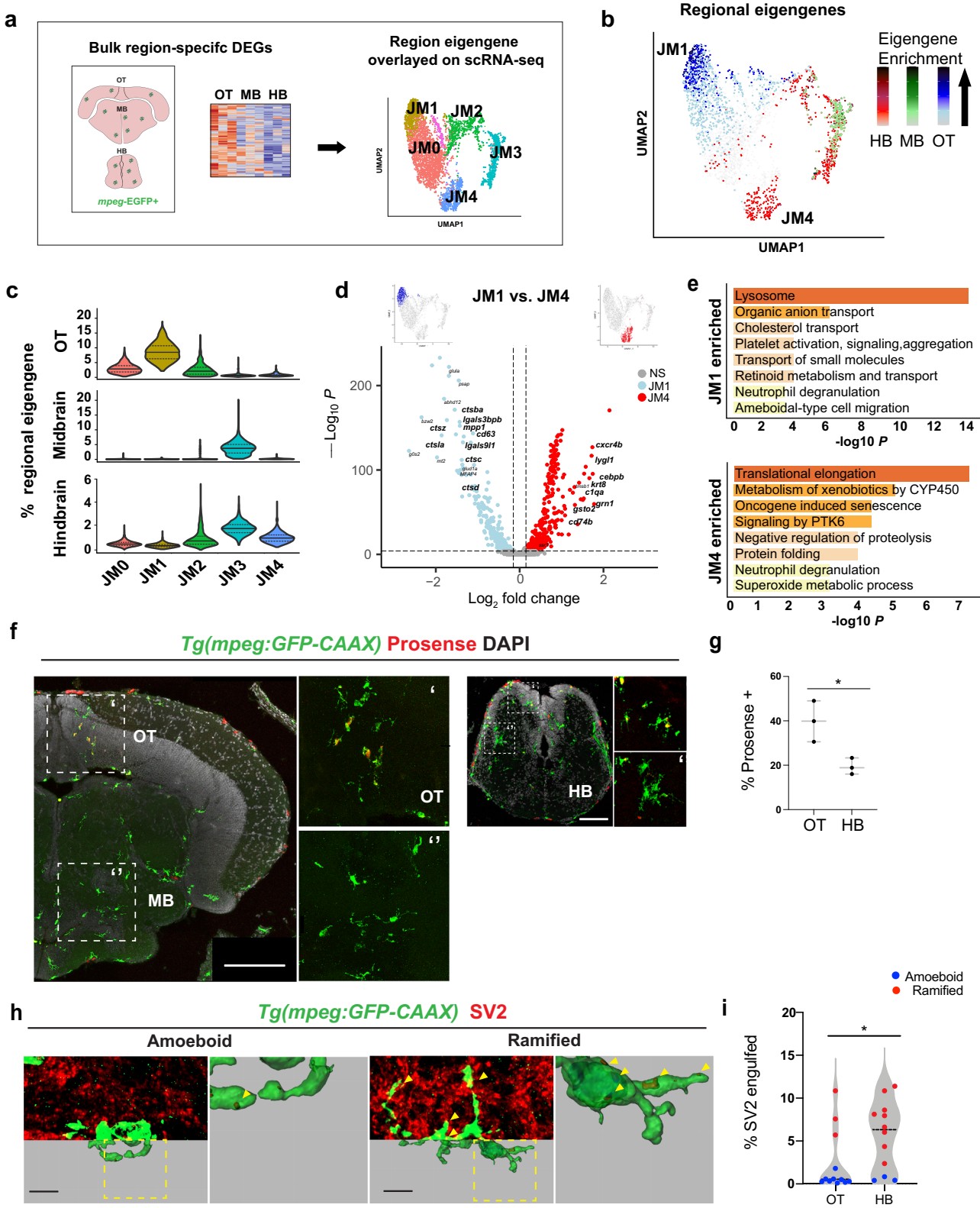

In contrast, the OT-enriched cluster JM1 was highly enriched for lysosomal activity, including multiple genes for lysosomal proteases including cathepsins (*ctsba, ctsz, ctsla, ctsc*, and *ctsd*) as well as higher levels of *apoeb* and *lgals9l1*. Thus, functional subsets identified by scRNA sequencing correlate with the differentially enriched populations of synapse-associated and neurogenic-associated microglia that we identified in situ.

We hypothesized that hindbrain-enriched cluster JM4 represents synapse-associated microglia (SAM) subset, whereas OT-enriched JM1 preferentially associates with neurogenic regions. To identify our putative OT-enriched cathepsin-rich cluster (JM1) in situ, we quantified functional cathepsin activity with the biomarker Prosense680, which becomes fluorescent after proteolytic cleavage by cathepsins with a preference for cathepsin B and

**Fig. 4 Region-enriched functional microglial subsets in zebrafish hindbrain and optic tectum. a** Schematic of analysis pipeline overlaying brain-region-defining genes identified with bulk-sequencing onto *mpeg1*+ scRNA-sequencing clusters. Colors correspond to clusters defined in Fig. 2c. **b** Feature plots of region-defining eigengenes from hindbrain (HB; red), midbrain (MB; green), and optic tectum (OT; blue) overlaid on *mpeg1*+ UMAP, highlighting microglial clusters JM1 and JM4. Eigengenes were composed of single region marker genes with regional enrichment ($\log_2$ fold change >1.2, basemean >100, and adj. $p < 0.05$) from the bulk-sequencing analysis in Fig. 3 and computed with the PercentageFeatureSet function in the Seurat R package. Color-coding for Eigengene expression: OT, low expression; light blue, high expression; dark blue. MB, low expression; light green, high expression; dark green. HB, low expression; pink, high expression; red. **c** Violin plots of regional eigengene distributions, related to **b**. Colors correspond with clusters shown in Figs. 2c, 4a. (Solid black line = median; dotted lines = 1st and 3rd quartiles). **d** Volcano plot of differentially expressed genes between clusters JM1 (OT-enriched; blue) and JM4 (hindbrain-enriched; red). Thresholds represented by dotted lines were set to adjusted $p$ value $< 10^{-8}$, log fold change >0.2. (MAST differential expression test with Bonferroni correction; See Supplemental S5). **e** Top GO terms from differentially expressed genes in **d**. (Metascape; hypergeometric test with Benjamini–Hochberg correction for multiple comparisons). **f** Representative images of cathepsin-cleaved Prosense680 colocalized with *mpeg1.1*-EGFP in indicated brain regions. Scale: 100 μm. Images are representative of the $n = 3$ replicates. **g** Quantification of total percent microglia containing cleaved Prosense680 in OT and HB. Unpaired $t$ tests. Dots represent three individual fish, data are mean ± SD. *$p < 0.0236$. OT (Optic Tectum) and HB (Hindbrain). Two-tailed unpaired $t$ test. **h** Representative images and 3D reconstructions of *mpeg1.1*-EGFP+ microglia with engulfed SV2 protein. Insets: close-up of reconstructions (arrowheads:SV2 content). Scale bar 5 μm. Images are representative of ameboid or ramified morphology. **i** Quantification of percent microglial volume containing SV2 in randomly selected microglia from OT and HB. Post hoc analyses show morphology assignment as ramified (red, sphericity <0.6) vs. ameboid (blue, sphericity >0.6. Mann–Whitney $U$ test, *$p < 0.0177$, dots = mean value per fish from three microglia per fish. See also Fig. S5 and Fig. S6, Supplementary Data S5.

L[45]. Microglial cathepsin activity (*mpeg1.1*-EGFP+ Prosense+) was highly enriched in the OT, where it was mostly detected in ameboid microglia around neurogenic regions (Fig. 4f, g), closely matching the expression of *ctsba* by ISH (Fig. 3f, g). Consistent with this, OT microglia engulfed substantially more cell corpses than hindbrain microglia, as quantified by uptake of a neuronal-soma enriched DsRed *Tg(NBT:DsRed)*; Fig. S6a, b). Next, we quantified synapse engulfment by 3D reconstruction of *mpeg1.1*-EGFP positive cells with the synaptic marker SV2. We found that hindbrain microglia engulfed significantly more SV2 than OT microglia, consistent with enrichment of the JM4 cluster in hindbrain (Fig. 4h, i). Importantly, stratifying this data by morphology revealed that ramified microglia (sphericity < 0.6) engulfed more SV2 regardless of a brain region, although they were markedly more abundant in the hindbrain. These data suggest functional subsets whose abundance differs between brain regions, rather than strictly region-specific functions. Taken together, we conclude that our cathepsin-enriched cluster JM1 corresponds to cell-corpse-engulfing microglia highly enriched in neurogenic regions, whereas complement-expressing cluster JM4 identified a synapse-associated subset.

We performed additional analyses to assess lineage relationships and determine whether these functional subsets were conserved across species. To examine predicted lineage relationships, we performed pseudotime analysis with Monocle 3[46–48]. This suggested that adult-specific cluster A4, NAMs, and SAMs each represent distinct endpoints derived from precursors that include the proliferating and putative homeostatic clusters A2/JM2 and A0/JM0 (Fig. S7a). To determine whether these subsets could be identified in the developing human brain, we compared the clusters to a published human fetal (HF) microglia dataset (Kracht et al. 2020, Fig. S7b)[49]. We found that 71% (750/1054) of the human cluster marker genes analyzed had at least one zebrafish homolog. Using these genes, we found a strong correlation between HF cluster 6 (dividing cells) and zebrafish cluster JM2 (dividing cells); additionally, HF clusters 3 (enriched in gestational weeks 11–12) and 5 (immediate early gene-expressing microglia) had significant overlap with NAM cluster JM1. Together, the NAM-overlapping clusters (3 and 5) represented 20% of cells. A smaller, less well-characterized cluster (<3% of cells) shared a gene signature with SAM cluster JM4. From this, we conclude that NAM-like cells can be identified in a HF data set, whereas SAMs are either rare at this developmental stage or less functionally discrete in the HF brain. Thus, these functional subsets of microglia have a

conserved gene signature that is at least to some extent detectable across species.

## Discussion

Here, we define the regional localization and molecular signatures of two distinct microglial phagocytic states in the developing zebrafish, including a subset of SAMs abundant in the hindbrain (Fig. 5a, b). We also identify several other immune populations, including a subset of macrophages. However, owing to the possible presence of circulating cells in the sample, and limited conservation between zebrafish and mammalian CNS macrophage markers, future targeted investigations of these non-microglial populations are warranted. Nonetheless, this functionally annotated microglial single-cell data set presents an opportunity to investigate fundamental questions related to interactions between microglia and synapses. For example, microglia both engulf synapses and promote synapse formation via modification of the extracellular matrix and other mechanisms[1,50,51], but the molecular regulators of these different states are not well understood. The impact of neuronal activity on microglial function is also a major area of interest: microglial–synapse engulfment has been proposed to be activity-dependent, and in both fish and rodents microglial contact can acutely regulate neuronal activity[52,53]. However, observing these processes in the intact developing brain is challenging in rodents and is a major strength of the zebrafish model. This molecularly defined population of SAM in the zebrafish hindbrain provides an opportunity to temporally define and genetically manipulate these microglial subsets in physiological and disease contexts.

Importantly, we show that both SAMs and NAMs are phagocytic in vivo, but in different contexts and via distinct molecular mechanisms. Our observations regarding NAMs are consistent with several recent publications. For example, deficits in lysosomal function lead to defective cell-corpse phagocytosis and dysmorphic "bubble microglia" in the OT[21]. Another study used bulk RNA sequencing to identify *ccl34b.1* as a marker of ameboid microglia with high phagocytic capacity in the adult zebrafish[32]. Interestingly, *ccl34b.1* is also strongly enriched in our NAM subset, suggesting that these represent similar or overlapping populations, and possibly that the ontogenetic origins of those microglia as described in that study may be linked to their distinct functions, although further work would be required to definitively establish this. Our studies of functional cathepsin activity suggest in vivo approaches to track lysosomal function in this population. More importantly, we have identified genes associated with microglia localized to synaptic regions of the zebrafish CNS. The

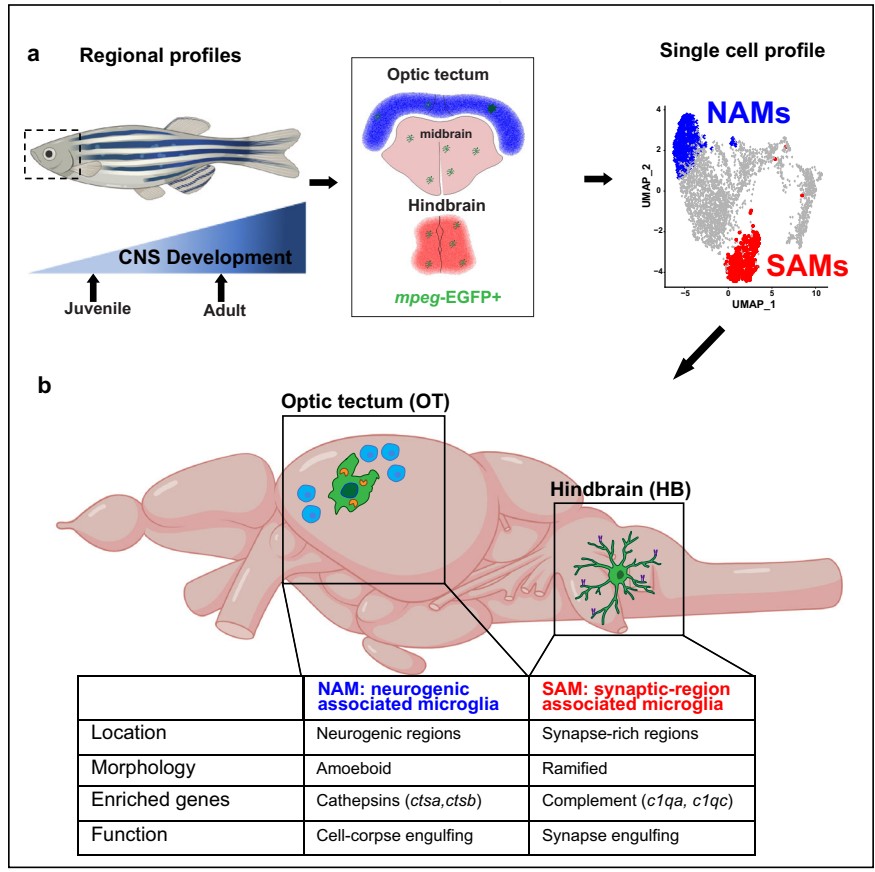

**Fig. 5 Summary diagram of the identified neurogenic-associated microglia (NAMs) and synaptic-region associated microglia (SAMs) in the developing zebrafish brain. a** Schematic of analysis pipeline overlaying brain-region-defining genes identified with bulk-sequencing onto *mpeg1.1*+ scRNA seq clusters identify NAMs and SAMs. **b** Unique features of NAMs and SAMs in the developing juvenile brain. Adapted from BioRender.com (2020). Retrieved from https://app.biorender.com/biorender-templates.

initiating components of the classical complement cascade (*c1qa* and *c1qc*) are top differentially expressed genes in our SAM subset. C1q is also microglial-encoded in the murine brain and in multiple vertebrate species including zebrafish[9,41] and promotes developmental synaptic engulfment[54]. Other top candidates, including the transcription factor *cebpb* and major histocompatibility complex, class II (*cd74a & cd74b*), have been implicated in rodent models of Alzheimer's disease[43,55]. These data suggest that the SAM profile identifies a core microglial program conserved across species, and raises the question of how other gene candidates, including *lygl1, grn1, fgl2a,* and others, are regulating microglial-synaptic interactions.

Our data highlight evolutionarily conserved features of zebrafish microglia as well as unique strengths of the zebrafish model. It is striking that developmental microglial heterogeneity in the fish is largely conserved into adulthood, in contrast to generally diminishing amounts of diversity in adult rodents[3,4]. This may reflect the fact that fish continue to grow in size throughout life, adding new neurons and new synaptic connections. This conservation suggests that the time window for studying microglial roles in circuit formation may be much broader than in mammals, and that mechanisms may exist to maintain synaptic plasticity throughout life. Future studies focusing on the microglial–synapse interactions in the zebrafish hindbrain can take advantage of co-existing populations of cell-corpse engulfing and SAM to further define the differences between these subsets. Our data also suggest that live-imaging of microglia-synapse interactions may help to answer key questions about how and why microglia interact with synapses. Finally, the ability to do

high throughput screening in zebrafish raises the possibility that this model could be used to define therapeutic targets in neuro-developmental diseases linked to immune dysfunction including autism spectrum disorder, epilepsy, and schizophrenia.

## Methods

**Experimental model and subject details**. All animal protocols were approved by and in accordance with the ethical guidelines established by the UCSF Institutional Animal Care and Use Committee and Laboratory Animal Resource Center (LARC). Wild-type, AB strain zebrafish (*Danio rerio*; ZIRC, University of Oregon, Eugene, OR) were propagated, maintained, and housed in recirculating habitats at 28.5 °C and on a 14/10-h light/dark cycle. Embryos were collected after natural spawns, incubated at 28.5 °C, and staged by hours post fertilization (hpf). Juveniles used were 28 days of age, a time in development before sex determination. Adults of either sex used were at 12 months of age. Ages were matched within experiments. The transgenic reporter lines, *Tg(cd45:DsRed)* and *Tg(mpeg1.1:EGFP)* was used to identify hematopoietic lineage and mononuclear phagocytes[33,56].

**Immunohistochemistry**. Tissues were fixed overnight at 4 °C in 0.1 M phosphate-buffered 4% paraformaldehyde, cryoprotected with 20% sucrose, and embedded in optimal cutting temperature (OCT) medium (Sakura Finetek USA, Torrance, CA). IHC was performed on 20-μm-thick sections collected on a cryostat and mounted onto glass slides[57]. Sections were washed in phosphate-buffered saline with 0.5% Triton-x (PBST) and incubated with 20% heat-inactivated normal goat serum in PBST for 2 h (NSS; Sigma-Aldrich, Corp.). Primary antibodies were applied overnight at 4 °C, all antibody dilutions and catalog numbers are listed in Supplemental Table 1. Sections were then washed with PBST and incubated in secondary antibodies for 1 h at room temperature. Prior to IHC for BrdU, sections were immersed in 100 °C sodium citrate buffer (10 mM sodium citrate, 0.05% Tween 20, pH 6.0) for 30 min and cooled at room temperature for 20 min. IHC was performed as described above.

**Labeling neurogenic zones**. In juveniles, dividing cells were labeled with BrdU by housing animals in system water containing 5 mM BrdU for 24 h prior to collection[57].

**In situ hybridization**. Digoxigenin (DIG)-labeled riboprobes for *cd74a* and *ctsba* was generated by PCR amplification using primers containing the T3 or T7 promoter sequences. Full sequences are available in Supplemental Table 1. In all, 20-µm-thick sections were hybridized with riboprobes at 55 °C, incubated with an alkaline-phosphatase-conjugated anti-DIG antibody, and visualized using Fast Red TR/Naphthol AS-MX (SIGMA*FAST*) as the enzymatic substrate. When ISHs were combined with BrdU IHC, sections were removed from the fast red solutions, rinsed, and post-fixed in buffered 4% paraformaldehyde for 10 min then processed for BrdU IHC as described above.

**ProSense680 injections**. Fish were injected with 2 nL of ProSense680 at a concentration of 20 nM using a nanospritzer with a fire pulled glass pipette connected directly into the lateral ventricle of the brain and returned to system water immediately following injections. Fish were euthanized 24 h post injection and processed for immunohistochemistry as stated above.

**Microglia morphology quantitation**. Z-stack images of *mpeg1.1*:EGFP + and 4C4 staining were acquired with a step size of 0.5 µm using a ×63 objective (NA 1.4) on an LSM 800 Confocal Microscope (Zeiss) spanning a thickness of 20 µm. Microglia sphericity and Sholl intersections were quantified using Imaris software (Bitplane) by creating 3D surface reconstructions of *mpeg1.1*-EGFP+-4C4+ microglia. All images were set to a standard threshold to accurately maintain morphology for quantifications.

**Microglia engulfment assay**. Images were acquired with an LSM 800 Confocal Microscope (Zeiss) using the same parameters as described above. Imaris software (Bitplane) was used to generate 3D surface rendering of microglia, which were then masked for *NBT*-DsRed or SV2 channels within that microglia. Masked channels were then 3D rendered to obtain volume data. *NBT*-DsRed and SV2 engulfment was calculated per cell as the volume of SV2 or *NBT*-DsRed divided by the volume of the microglia.

**Fluorescence-activated cell sorting (FACS)**. For bulk RNA-sequencing of juvenile 28 dpf *Tg(mpeg1.1:EGFP)* zebrafish, the OT, midbrain, and hindbrain were dissected (10 zebrafish were pooled per sample). For scRNA-sequencing of 28 dpf *Tg(mpeg1.1:EGFP): Tg(cd45:DsRed)* juveniles (10 zebrafish pooled per lane) and 1-year old *Tg(mpeg1.1:EGFP): Tg(cd45:DsRed)* adults (3 zebrafish pooled per lane) whole brains were dissected. To isolate microglia and other *cd45* + cells, the brain(s) (regions) were mechanically dissociated in isolation medium (1× HBSS, 0.6% glucose, 15 mM HEPES, 1 mM EDTA pH 8.0) using a glass tissue homogenizer (VWR). Subsequently, the cell suspension was filtered through a 70 µm filter (Falcon) and pelleted at 300 g, 4ºC for 10 min. The pellet was resuspended in 22% Percoll (GE Healthcare) and centrifuged at 900 × g, 4ºC for 20 min (acceleration set to 4 and deceleration set to 1). Afterward, the myelin-free pellet was resuspended in an isolation medium that did not contain phenol red. Prior to sorting on a BD FACS Aria III, the cell suspension was incubated with DAPI (Sigma). For bulk RNA-sequencing, microglia were gated on FSC/SSC scatter, live cells by DAPI, and *mpeg1.1*:EGFP+. After sorting, cells were spun down at 500 × g, 4ºC for 10 min, and the pellet was lysed with RLT + (Qiagen). For scRNA-sequencing, microglia, macrophages, and *cd45* + cells were collected by gating on FSC/SSC scatter, live cells by DAPI, and all *cd45*:DsRed (which included both *mpeg1.1*:EGFP + and negative subsets). After sorting, cells were spun down at 500 × g, 4 °C for 10 min and resuspended in PBS + 0.05% bovine serum albumin (Sigma).

**Bulk RNA-sequencing of microglia**. RNA was extracted using the RNeasy® Plus Micro kit (Qiagen) from RLT+ lysed microglia. RNA quality and concentration were measured using the Agilent RNA 6000 Pico kit on an Agilent Bioanalyzer. All samples had an RNA integrity number > 8. For each sample, a total of 10 ng of RNA was loaded as input for cDNA amplification and library construction using the QuantSeq 3' mRNA-Seq Library Prep Kit FWD for Illumina (Lexogen) following the manufacturer's instructions. Library quality was determined with the Agilent High Sensitivity DNA kit on an Agilent Bioanalyzer and concentrations measured with the Qubit™ dsDNA HS Assay Kit (Thermo Fisher) on a Qubit™ (Thermo Fisher). Library pools were single-end (65-bp reads) sequenced on two lanes using an Illumina HiSeq 4000 yielding 40-50 million reads per sample.

**Bulk RNA-sequencing analysis**. Quality of reads was assessed using FastQC (http://www.bioinformatics.babraham.ac.uk/projects/fastqc). All samples passed quality control and reads were aligned to *Danio* rerio GRCz11.98 genome (retrieved from Ensemble) using STAR (version 2.5.4b)[58] with "--outFilterMultimapNmax 1" to only keep uniquely mapped reads. Uniquely mapped reads were counted using HTSeq (version 0.11.1)[59]. Subsequently, the DESeq2 package (version 1.28.1)[60] in R software was used to normalize the raw counts and perform differential gene expression analysis. The batch correction was done using the Limma package (version 3.44.3)[61] and heatmaps were made using ComplexHeatmap package (version 2.4.3)[62]. Metascape was used for gene ontology analysis[63].

**Single-cell RNA-sequencing**. Single cells were isolated as described above. Approximately 15,000 cells were loaded into each well of Chromium Chip B (v3), libraries were prepared using the 10× Genomic Chromium 3' Gene Expression Kit in-house as described in the Manual and sequenced on two lanes of the NovaSeq SP100 sequencer for an average depth of 30,000–50,000 reads per cell. BCL files are converted to Fastq, then used as inputs to the 10× Genomics Cell Ranger 2.1 pipeline. Samples were aligned to the GRCz11.94 (danRer11) zebrafish reference genome. Clustering and differential expression analysis were conducted using Seurat version 3.1.4[64,65]. Cells outside of the thresholds listed in the table were excluded from downstream analysis.

Feature thresholds

| | Whole-brain juvenile isolation | Whole-brain adult isolation |
|---|---|---|
| Number of cells | Loaded: 20,000 Total passing QC: 6666 *mpeg1.1*:EGFP+: 3539 | Loaded: 10,000 *mpeg1.1*:EGFP + after QC: 2336 |
| Biological replicates | 10 fish | 3 fish |
| 10× lanes | 2 | 1 |
| Age | 28 days post fertilization | 1-year post fertilization |
| 10× chip kit | V3 | V3 |
| Thresholds | 500–3500 genes/cell; 1200–15,000 transcripts/cell; 0-10% mitochondrial transcripts | 500–3500 genes/cell; 1200–15,000 transcripts/cell; 0–10% mitochondrial transcripts |

Counts were log normalized (scale factor = 10,000) and scaled in Seurat, regressing out a number of genes detected (nFeature_RNA). The top 6000 most variable genes (calculated with the vst method in Seurat) were used to calculate 50 principal components, and the top 30 PCs were used for the nearest neighbor, UMAP, and cluster calculations with the resolutions shown in the table. Individual cell types were identified through calculation of marker genes using the MAST test (version 1.16.0)[66] for genes expressed in at least 20% of cells in the cluster and a natural log fold change of 0.2 or greater and adjusted $p$ value < $10^{-8}$. The microglial and macrophage clusters were isolated based on the expression of *cd45* (*ptprc*) and *mpeg1.1*. Normalization, clustering, and differential gene expression were recalculated for each sample (juvenile, juvenile *mpeg1.1*:EGFP+, adult, and juvenile *mpeg1.1*:EGFP+) on genes expressed in 10% or more cells per cluster. $P$ values were calculated for genes with a natural log fold change >0.2 and genes with an adjusted $p$ value < 0.001 were used for further analysis. GO analysis was conducted using the Metascape webpage (www.metascape.org)[63]. Adult and juvenile datasets were combined using Harmony (version 1.0)[40]. Bulk vs single-cell analysis was conducted using the following thresholds for bulk-sequencing results: adj. $p < 0.05$, $log_2$ fold change >1.2, basemean >100. Following thresholding, only genes uniquely enriched in one region were used to calculate "eigengene" values determined with Seurat's "PercentageFeatureSet" function.

Psuedotime analysis was conducted with Monocle 3 (version 1.0.0)[46–48] with batch correction for age and regression of percent mitochondria and total RNA counts per cell using Batchelor (version 1.6.3)[67]. All other default parameters were used, and dividing cells were selected as the starting point for pseudotime calculations.

Comparison with HF microglia[49] was conducted by converting gene IDs from the differentially expressed genes per HF cluster calculated by the authors in table S3 (average log fold change >0.25, adjusted $p$ value <0.05) into homologous zebrafish genes using biomaRt (version 2.46.3). In all, 354 of 1054 human genes were un-annotated and excluded from the calculation. The converted gene IDs were then used to calculate an eigengene/module score (Seurat's AddModuleScore function) for each HF cluster on all juvenile *mpeg1.1*+ microglia. Correlations between HF and JM clusters were conducted using a two-sided Wilcoxon rank-sum test for each HF cluster compared with all JM clusters 0–5. The test estimate for each comparison was used to create a heatmap, and asterisk labels represent associated $p$ values for the comparison of each specific JM cluster's enrichment in HF cluster score compared with all other JM clusters.

**Quantification and statistical analysis**. Graphpad Prism 8.3.0 was used for all histological quantification analyses. Statistical tests are described in text and figure legends. RNA-sequencing data were analyzed in R as described above. Two-sided unpaired $t$ tests were used in comparing two groups in which the data were normally distributed, data sets with more than two groups were analyzed with one-way or two-way repeated-measures analysis of variance as appropriate.

## Data and materials availability

All data supporting the findings of this study are provided within the paper and its supplementary information. All additional information will be made available upon reasonable request to the authors. The single-cell RNA sequencing and bulk RNA-

sequencing data generated in this study have been deposited in the NCBI Gene Expression Omnibus database under accession code GSE164772 and GSE164771, respectively. The data are unrestricted. The processed data are available in Supplementary Information, and a searchable database is provided at https://www.annamolofskylab.org/microglia-sequencing. The human fetal microglia data analyzed in this study were downloaded from Table S3 of a previously published article[49] [https://science.sciencemag.org/highwire/filestream/747948/field_highwire_adjunct_files/0/aba5906_Kracht_SM.pdf]. Any additional information required to reanalyze the data reported in this paper is available from the lead contact upon request. Source data are provided with this paper.

## Code availability

All code used to analyze sequencing data can be found at https://github.com/lcdorman/zebrafish2021. Source data are provided with this paper.

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

## Acknowledgements
We are grateful to Marci Rosenberg, Haruna Nakajo, and members of the Molofsky Lab for helpful comments on the manuscript, to Ari Molofsky, Tom Nowakowski, and Galina Schmunk for their expert advice, and to Brian Black, Gary Moulder, and Louie Ramos for the support of the CVRI Zebrafish Core facility. Thanks to Francesca Peri for the Tg(*mpeg1.1:GFP-CAAX*) fish, David Traver for the *Tg(cd45:DsRed)* fish, and Roland Wu for the *Tg(mpeg1.1:EGFP)* fish, and to the UCSF Laboratory for Cell Analysis and Dr. Eric Chow at Center for Advanced Technologies for technical contributions. A.V.M is supported by the Pew Charitable Trusts, NIMH (R01MH119349 and DP2MH116507), and the Burroughs Welcome Fund. N.J.S. is supported by UCSF-IRACDA (K12GM081266) fellowship. L.C.D. received support from the Matilda Edlund Scholarship and the Genentech Fellowship.

## Author contributions
Conceptualization: N.J.S., A.V.M., I.D.V., and L.C.D.; methodology, N.J.S., I.D.V, and L.C.D.; investigation, N.J.S, I.D.V, L.C.D, and N.C.H.; writing—original draft, N.J.S., L.C.D., and A.V.M.; writing—review & editing, all co-authors; funding acquisition, A.V.M., and N.J.S. resources, A.V.M.; supervision, A.V.M. and I.D.V.

## Competing interests
The authors declare no competing interests.
