## [Peer Review File · Nature Communications]

Reviewers' Comments:

Reviewer #1:

Remarks to the Author:

Cell lineage studies, functional investigations and RNA-sequencing approaches have begun to uncover heterogeneity among microglia between brain regions and across developmental time and aging. The extent of this heterogeneity, how it is acquired and how it affects brain development and homeostasis remain as very important questions in the field. This manuscript advances our understanding of microglia heterogeneity by describing two distinct populations of microglia in the zebrafish brain. Although the work does not reveal how these microglia become different nor their specific roles in brain development and function, nevertheless I think this manuscript is an important contribution to the literature. In particular, the gene expression data, which I think is of high quality, will be a very rich resource for the field. For the most part, I think the data adequately support the claims made by the authors, that the statistical analyses are appropriate and that the methods and reagents are clearly described. My specific comments are relatively minor and mostly aim to increase clarity.

- Lines 68-69 – “highly homologous”. I realize this is very picky but homology is not quantitative. “Homologous” or “highly similar” would be more appropriate.
- Lines 74-75 – I think the sentence is missing a word.
- Lines 79-80 – “whether synapse-associated microglia also exist in the fish”. The way this is written is a little confusing, because the premise for synapse-associated microglia is not introduced prior to this statement.
- Lines 81-83 – “Bulk transcriptomic sequencing in the fish has begun to uncover key information regarding microglial ontogeny, revealing that microglia populate the CNS in two waves 29,30.” This is not an accurate statement. The evidence reported in the cited publications is based on fate mapping and lineage tracing strategies, not sequencing.
- Lines 111-112 – the authors state that they found microglia within synapse-rich regions “as early as 7 dpf”. I believe this is the earliest timepoint that the authors tested. If so, they should make that clear to avoid the possibility that people might interpret this statement to mean that none were evident prior to 7 dpf.
- I’m a little uncomfortable with the designation “synapse-associated microglia”. I understand that it is a convenient way to describe them following from the approaches in this manuscript but I worry that this description might, ultimately, be too narrow in terms of their cellular associations and functions. Perhaps the authors could consider another descriptor.
- Lines 117-119 and Figure S1 – The authors label mpeg:EGFP fish with a 4C4 antibody and use the results to conclude that a majority of mpeg:EGFP cells in the CNS are microglia. The implication in the text is that the 4C4-negative cells are macrophages and, indeed, this is stated in the Figure S1 heading. I think this point needs some clarification to avoid confusion. First, I’m not convinced that the 4C4 antibody is a definitive microglia marker, so I’m not convinced that a CNS parenchymal mpeg:EGFP-positive 4C4-negative cell is not a microglia. Second, it is not apparent to me which cells the authors are referring to. Are these mpeg:EGFP-positive 4C4-negative cells in the parenchyma, or are they non-parenchymal, perivascular or meningeal macrophages? No less an authority than Ben Barres has said that all parenchymal macrophages are microglia, so I just don’t want people to come away from this manuscript thinking that there is yet another class of brain macrophage.
- The previous point also is relevant to the scRNA-seq cell clustering analysis, from which the authors identify a macrophage (JM3) cluster. I think it is important to know if these are likely non-parenchymal macrophages.
- I would encourage the authors to double check their genetic nomenclature for consistency (e.g. mpeg:EGFP, mpeg:eGFP, mpeg-EGFP; 4C4, 4c4) and for conformity to guidelines available at ZFIN (e.g. mpeg should be mpeg1.1).
- Wu et al 2020 argue that ameboid microglia are derived both from the rostral blood island (RBI) and the aorta-gonad-mesonephros (AGM) whereas ramified microglia are entirely from the AGM in zebrafish. I think this potentially provides some insight to mechanisms that determine the phenotypic and functional differences of the neurogenic-associated and synapse-associated microglia that these authors describe. It might be worth commenting on this in the Discussion.

Bruce Appel

Reviewer #2:

Remarks to the Author:

Silva et al present an elegant study analysing different subsets of microglia in larval and adult zebrafish brains. They combine single cell sequencing, regional bulk sequencing, immuno histochemistry and in situ hybridisation to analyse the variety of microglia. Based on this, they identify different populations and highlight two major populations, synapse associated microglia and neurogenic associated microglia.

Although rather descriptive, this study is in my opinion highly relevant for the zebrafish field as it provides a detailed assessment of microglia sub populations which is clearly needed for future mechanistic studies.

I did not identify any technical issues, the study is sound and all conclusions are backed up by strong data. My only recommendation would be to adjust Figure S1: provide images for the single channels and provide sufficient labelling (mpeg vs 4C4).

Reviewer #3:

Remarks to the Author:

Silva and co-workers describe 2 subsets of microglia in the developing zebrafish brain using scRNAseq. The paper further contributes to our understanding of microglia subsets in the developing CNS. Here a phagocytic, neuronal-corpse engulfing subtype in the optic tectum and a synapse-associated subtype in the hindbrain are reported.

The strength of the paper is that the functional properties of these microglia subsets are confirmed using Prosense for proteolytic activity and quantifying SV content for synapse engulfment. A weakness is the low number of cells subjected to scRNAseq, somewhat hampering the interpretation of the scRNAseq data, which is inherently less sensitive than bulk RNAseq, which in its turn lacks cellular resolution. For more robust clustering and marker genes etc, more cells would help.

The paper reads well and the data are clearly presented, several points for improvements are listed

1 Last year, a paper reporting on microglia heterogeneity in the human fetal brain was published in Science, using scRNAseq. The authors should determine the (dis)similarity of the NAMS and SAMS to the (phagocytic) microglia clusters in the human fetal CNS to compare microglia development in the zebrafish and human brains are.

2 In fig 1A, the DAPI signal is oversaturated, and individual nuclei cannot be distinguished. This makes it a bit difficult to interpret the % of mpeg-expressing microglia/total cell number.

3 In fig 1G, 2 "typical" microglia are depicted, more convincing would be, to depict the microglia located at the center/middle of the violins depicted in 1H. In 1C, fairly ramified cells can be seen in the OT.

4 The morphometrics depicted in 1H is a bit minimalistic, there are several tools available to perform a more sophisticated microglia morphology quantitation. Is microglia density altered (cells/surface), ramifications, etc. Based on the violin, there seem to be 2 microglia subsets in the OT, one centered around 0.6, the other around 0.8 sparsity.

5 In Fig 2, juvenile and adult microglia scRNAseq data are depicted. I assume, its not 100% clear from the text, that the 2 datasets were merged prior to the subclustering depicted in 2E.

6 Pseudotime analysis should be performed to delineate potential developmental trajectories of these microglia subsets in juvenile and adult fish.

7 Overall, the number of microglia in the scRNAseq data set is not very high, 3529. This might preclude a more robust identification of subcluster marker genes. JMO, the largest cluster, did not have a regional enrichment signature. If it is indeed a common microglia subset, as proposed by the authors, it is difficult to imagine why it was not represented in the bulk RNAseq data. In 2C, cells from clusters 0 and 1 are somewhat mixed, were the data possibly over/underclustered or is

the # of cells too low for better separation?

July 27, 2021

Re: *In situ* and transcriptomic identification of microglia in synapse-rich regions of the developing zebrafish brain (NCOMMS-21-16792)

We sincerely thank all three reviewers for their detailed feedback, which we have carefully considered in revising our manuscript. This revision contains new data, including further characterization of 4C4 negative cells and morphologic analysis of microglia. We have also added several new bioinformatic analyses to extend our data, including comparison with a published human fetal dataset. As suggested by reviewers, the text has been corrected for clarity and consistency. We hope that in its revised form, this manuscript will be a useful resource to the zebrafish community and prompt further investigations of microglial-synapse interactions in this unique model organism.

A point-by-point response to each reviewer's comments follows below.

Reviewer 1: pp. 2-5

Reviewer 2: p. 6

Reviewer 3: pp. 7-11

Reviewer #1 (Remarks to the Author):

Cell lineage studies, functional investigations and RNA-sequencing approaches have begun to uncover heterogeneity among microglia between brain regions and across developmental time and aging. The extent of this heterogeneity, how it is acquired and how it affects brain development and homeostasis remain as very important questions in the field. This manuscript advances our understanding of microglia heterogeneity by describing two distinct populations of microglia in the zebrafish brain. Although the work does not reveal how these microglia become different nor their specific roles in brain development and function, nevertheless I think this manuscript is an important contribution to the literature. In particular, the gene expression data, which I think is of high quality, will be a very rich resource for the field. For the most part, I think the data adequately support the claims made by the authors, that the statistical analyses are appropriate and that the methods and reagents are clearly described. My specific comments are relatively minor and mostly aim to increase clarity.

- Lines 68-69 – “highly homologous”. I realize this is very picky but homology is not quantitative. “Homologous” or “highly similar” would be more appropriate.
- Lines 74-75 – I think the sentence is missing a word.
- Lines 79-80 – “whether synapse-associated microglia also exist in the fish”. The way this is written is a little confusing, because the premise for synapse-associated microglia is not introduced prior to this statement.
- Lines 81-83 – “Bulk transcriptomic sequencing in the fish has begun to uncover key information regarding microglial ontogeny, revealing that microglia populate the CNS in two waves 29,30.” This is not an accurate statement. The evidence reported in the cited publications is based on fate mapping and lineage tracing strategies, not sequencing.
- Lines 111-112 – the authors state that they found microglia within synapse-rich regions “as early as 7 dpf”. I believe this is the earliest timepoint that the authors tested. If so, they should make that clear to avoid the possibility that people might interpret this statement to mean that none were evident prior to 7 dpf.
- I’m a little uncomfortable with the designation “synapse-associated microglia”. I understand that it is a convenient way to describe them following from the approaches in this manuscript but I worry that this description might, ultimately, be too narrow in terms of their cellular associations and functions. Perhaps the authors could consider another descriptor.
- Lines 117-119 and Figure S1 – The authors label mpeg:EGFP fish with a 4C4 antibody and use the results to conclude that a majority of mpeg:EGFP cells in the CNS are microglia. The implication in the text is that the 4C4-negative cells are macrophages and, indeed, this is stated in the Figure S1 heading. I think this point needs some clarification to avoid confusion. First, I’m not convinced that the 4C4 antibody is a definitive microglia marker, so I’m not convinced that a CNS parenchymal mpeg:EGFP-positive 4C4-negative cell is not a microglia. Second, it is not apparent to me which cells the authors are referring to. Are these mpeg:EGFP-positive 4C4-negative cells in the parenchyma, or are they non-parenchymal, perivascular or meningeal macrophages? No less an authority than Ben Barres has said that all parenchymal macrophages are microglia, so I just don’t want people to come away from this manuscript thinking that there is yet another class of brain macrophage.
- The previous point also is relevant to the scRNA-seq cell clustering analysis, from which the authors identify a macrophage (JM3) cluster. I think it is important to know if these are likely non-parenchymal macrophages.
- I would encourage the authors to double check their genetic nomenclature for consistency (e.g. mpeg:EGFP, mpeg:eGFP, mpeg-EGFP; 4C4, 4c4) and for conformity to guidelines available at ZFIN (e.g. mpeg should be mpeg1.1).
- Wu et al 2020 argue that ameboid microglia are derived both from the rostral blood island (RBI) and the aorta-gonad-mesonephros (AGM) whereas ramified microglia are entirely from the AGM in zebrafish. I think this potentially provides some insight to mechanisms that determine the phenotypic and functional differences of the neurogenic-associated and synapse-associated microglia that these authors describe. It might be worth commenting on this in the Discussion.

Bruce Appel

Reviewer #1, response: We sincerely appreciate Dr. Appel’s detailed feedback, and the comment that “this manuscript is an important contribution to the literature. In particular, the gene expression data, which I think is of high quality, will be a very rich resource for the field.” We hope that as revised, this manuscript will be a useful resource and a baseline for future mechanistic studies. We address each point below:

1. Lines 117-119 and Figure S1 – The authors label *mpeg:EGFP* fish with a 4C4 antibody and use the results to conclude that a majority of *mpeg:EGFP* cells in the CNS are microglia. The implication in the text is that the 4C4-negative cells are macrophages and, indeed, this is stated in the Figure S1 heading. I think this point needs some clarification to avoid confusion. First, I’m not convinced that the 4C4 antibody is a definitive microglia marker, so I’m not convinced that a CNS parenchymal *mpeg:EGFP*-positive 4C4-negative cell is not a microglia. Second, it is not apparent to me which cells the authors are referring to. Are these *mpeg:EGFP*-positive 4C4-negative cells in the parenchyma, or are they non-parenchymal, perivascular or meningeal macrophages? No less an authority than Ben Barres has said that all parenchymal macrophages are microglia, so I just don’t want people to come away from this manuscript thinking that there is yet another class of brain macrophage.

Response: We agree that this should have been more precisely addressed in our manuscript and that the title of Fig. S1 was misleading. We now include new data characterizing *mpeg*-GFP+ 4C4^{neg} cells by their

morphology and location relative to vessels and the brain borders (**Reviewer Fig. 1**). As expected, border associated macrophages (BAMs) as defined by location and elongated morphology, are all 4C4 negative. These comprise about 25% of 4C4^{neg} cells. The remaining 4C4^{neg} cells are indeed parenchymal. We separately quantified parenchymal cells contacting *flk1*-mCherry+ vessels (‘parenchymal, vessel contacting’) and 4C4^{neg} cells that were not vessel contacting. Of note, we did not observe morphologically distinct perivascular macrophages as we and others have observed in rodents – elongated cells thought to be bounded on both sides by basement membrane (Prinz et al. 2021). The vessel contacting cells we observe are ramified and look indistinguishable from microglia in their morphology. As such, we conclude that ~75% of 4C4^{neg} cells are parenchymal and are likely microglia, although we cannot rule out that some may be perivascular macrophages. We have clarified the text and the title of Fig. S1 to make it clear that 4C4 is specific, but not entirely sensitive.

- Prinz, M., Masuda, T., Wheeler, M.A., and Quintana, F.J. (2021). Microglia and Central Nervous System-Associated Macrophages – from Origin to Disease Modulation. *Annu. Rev. Immunol.* 39, 251–277.

2. The previous point also is relevant to the scRNA-seq cell clustering analysis, from which the authors identify a macrophage (JM3) cluster. I think it is important to know if these are likely non-parenchymal macrophages.

Response: This is an important point. We have added additional supplementary analyses and additional discussion to the text, clarifying that more targeted analyses will be required to definitively identify this macrophage subset (**Reviewer Fig. 2**). Briefly, cluster JM3 is consistent with macrophages (lack of *p2ry12*, *hexb*, and *csf1ra*) and could potentially include perivascular, meningeal, or circulating macrophages, as these animals were not perfused and the meninges were not removed. However, of the recently proposed mammalian BAM markers with fish homologs, none clearly segregated as expected and in some cases were not detected at all. These included mammalian genes *CD163*, *Mrc1*, *Apoc1*, *Apoc4*, *Lyve1* (Utz et al 2020, Van Hove et al 2019) of which the fish homologs, respectively, are *cd63*, *mrc1b*, *apoc1*, *apoc4*, and *lyve1a/b*. However, as demonstrated in situ, these cells are clearly present. Either too few were recovered to be resolved by scSeq, or fish BAMs at this age differ significantly from mammalian BAMs in their gene expression profile. Known canonical mammalian monocyte-derived macrophage markers that might define circulating vs. tissue resident macrophages are also not well annotated in zebrafish and/or not detected in our dataset (e.g. *Ly6c1*, *F4/80/adgre10*). As such, we cannot definitively identify these macrophages based on existing datasets. We now note this in the discussion and propose as a future

direction targeted examination of these subsets, which might be best accomplished by bulk RNAseq of purified microdissected meningeal macrophages vs. microglia.

- Utz, S.G., See, P., Mildenerger, W., Thion, M.S., Silvin, A., Lutz, M., Ingelfinger, F., Rayan, N.A., Lelios, I., Buttgerit, A., et al. (2020). Early Fate Defines Microglia and Non-parenchymal Brain Macrophage Development. *Cell* 181, 557-573.e18.
- Van Hove, H., Martens, L., Scheyltjens, I., De Vlaminc, K., Pombo Antunes, A.R., De Prijck, S., Vandamme, N., De Schepper, S., Van Isterdael, G., Scott, C.L., et al. (2019). A single-cell atlas of mouse brain macrophages reveals unique transcriptional identities shaped by ontogeny and tissue environment. *Nat. Neurosci.* 22, 1021–1035.

The remaining points are addressed in chronological order:

• Lines 68-69 – “highly homologous”. I realize this is very picky but homology is not quantitative. “Homologous” or “highly similar” would be more appropriate.

Response: Thank you for the suggestion, we have changed this to “homologous”.

• Lines 74-75 – I think the sentence is missing a word.

Response: Corrected, thank you.

• Lines 79-80 – “whether synapse-associated microglia also exist in the fish”. The way this is written is a little confusing, because the premise for synapse-associated microglia is not introduced prior to this statement.

Response: We agree. This section has been substantially re-written in response to this and subsequent comments, and we hope the logic is now clearer.

• Lines 81-83 – “Bulk transcriptomic sequencing in the fish has begun to uncover key information regarding microglial ontogeny, revealing that microglia populate the CNS in two waves 29,30.” This is not an accurate statement. The evidence reported in the cited publications is based on fate mapping and lineage tracing strategies, not sequencing.

Response: Thank you for this correction, we have now re-written this section to clarify the relationship between ontogeny and function (see reviewer point below), and have updated our citations in a manner that we hope will be more accurate.

• Lines 111-112 – the authors state that they found microglia within synapse-rich regions “as early as 7 dpf”. I believe this is the earliest timepoint that the authors tested. If so, they should make that clear to avoid the possibility that people might interpret this statement to mean that none were evident prior to 7 dpf.

Response: This statement has been corrected.

• I’m a little uncomfortable with the designation “synapse-associated microglia”. I understand that it is a convenient way to describe them following from the approaches in this manuscript but I worry that this description might, ultimately, be too narrow in terms of their cellular associations and functions. Perhaps the authors could consider another descriptor.

Response: Thank you for this suggestion. We appreciate the risk of overinterpretation when naming these subsets, while also recognizing the utility of a nomenclature. We have changed the descriptor to “synaptic-region associated microglia” to focus on location rather than function, but preserved the acronym SAM for simplicity. The title has also been revised to read: “*In situ* and transcriptomic identification of microglia in synapse-rich regions of the developing zebrafish brain”.

• I would encourage the authors to double check their genetic nomenclature for consistency (e.g. *mpeg:EGFP*, *mpeg:eGFP*, *mpeg-EGFP*; *4C4*, *4C4*) and for conformity to guidelines available at ZFIN (e.g. *mpeg* should be *mpeg1.1*).

Response: Apologies for this oversight, all genetic nomenclature was reviewed and corrected for consistency.

Wu et al 2020 argue that ameboid microglia are derived both from the rostral blood island (RBI) and the aorta-gonad-mesonephros (AGM) whereas ramified microglia are entirely from the AGM in zebrafish. I think this potentially provides some insight to mechanisms that determine the phenotypic and functional differences of the neurogenic-associated and synapse-associated microglia that these authors describe. It might be worth commenting on this in the Discussion.

Response: Thank you for this comment. Wu et al identified *cc134b.1* as a marker of a subset of ameboid phagocytic microglia derived from the rostral blood island (RBI) and aorta-gonad-mesonephros (AGM). This marker is also a high confidence hit in our NAM signature, suggesting that NAMs are very likely similar to the subset they identified, and derived from RBI/AGM. It is reasonable to assume that SAMs represent a subset of the ramified, AGM-derived, *cc134b.1*-negative cells described by Wu et al., although we also identify several additional clusters that they were not able to resolve by bulk-sequencing. A link between ontogeny and function may exist, but would require further investigation. We have now added this topic to our discussion.

Reviewer #2 (Remarks to the Author):

Silva et al present an elegant study analysing different subsets of microglia in larval and adult zebrafish brains. They combine single cell sequencing, regional bulk sequencing, immuno histochemistry and in situ hybridisation to analyse the variety of microglia. Based on this, they identify different populations and highlight two major populations, synapse associated microglia and neurogenic associated microglia. Although rather descriptive, this study is in my opinion highly relevant for the zebrafish field as it provides a detailed assessment of microglia sub populations which is clearly needed for future mechanistic studies. I did not identify any technical issues, the study is sound and all conclusions are backed up by strong data. My only recommendation would be to adjust Figure S1: provide images for the single channels and provide sufficient labelling (mpeg vs 4C4).

Reviewer #2, response: We thank the reviewer for their positive comments and hope that this manuscript will indeed help to promote future mechanistic studies.

My only recommendation would be to adjust Figure S1: provide images for the single channels and provide sufficient labelling (mpeg vs 4C4)

Response: Thank you for this comment, we now present single channel images (Reviewer 2, Fig. 3). In addition, we have performed additional characterization to better define the identity of 4C4 negative cells, which we hope will be informative. All of these data are now in manuscript Figure S1.

Reviewer 2 Fig. 3 (from Manuscript Fig. S1): *mpeg:GFP-CAAX* and 4C4 positive cells in zebrafish optic tectum and midbrain. (A) Representative images now include single channel insets illustrating colocalization *mpeg-GFP* and 4C4 for each respective region (B) quantification. C-E (New analysis): Further characterization of 4C4 negative cells in zebrafish optic tectum, midbrain, and hindbrain. C-D) Representative images of *Tg(mpeg-GFP-CAAX)* crossed with *Tg(flk1:mcherry)* labeling vasculature, stained with 4C4. 4C4-negative *mpeg-GFP-CAAX* positive cells were grouped into three categories: Border-associated macrophages (BAMs, i), vessel-contacting parenchymal cells (ii), and non-vessel contacting parenchymal cells (iii). Arrowheads indicate 4C4 negative/*mpeg-GFP-CAAX* positive cells. (E) Quantifications of the identity of 4C4 negative *mpeg-GFP-CAAX* positive cells in each region (n=4 fish/group).

Reviewer #3 (Remarks to the Author):

Silva and co-workers describe 2 subsets of microglia in the developing zebrafish brain using scRNAseq. The paper further contributes to our understanding of microglia subsets in the developing CNS. Here a phagocytic, neuronal-corpse engulfing subtype in the optic tectum and a synapse-associated subtype in the hindbrain are reported.

The strength of the paper is that the functional properties of these microglia subsets are confirmed using Prosense for proteolytic activity and quantifying SV content for synapse engulfment. A weakness is the low number of cells subjected to scRNAseq, somewhat hampering the interpretation of the scRNAseq data, which is inherently less sensitive than bulk RNAseq, which in its turn lacks cellular resolution. For more robust clustering and marker genes etc, more cells would help.

The paper reads well and the data are clearly presented, several points for improvements are listed

1 Last year, a paper reporting on microglia heterogeneity in the human fetal brain was published in Science, using scRNAseq. The authors should determine the (dis)similarity of the NAMs and SAMS to the (phagocytic) microglia clusters in the human fetal CNS to compare microglia development in the zebrafish and human brains are.

2 In fig 1A, the DAPI signal is oversaturated, and individual nuclei cannot be distinguished. This makes it a bit difficult to interpret the % of mpeg-expressing microglia/total cell number.

3 In fig 1G, 2 "typical" microglia are depicted, more convincing would be, to depict the microglia located at the center/middle of the violins depicted in 1H. In 1C, fairly ramified cells can be seen in the OT.

4 The morphometrics depicted in 1H is a bit minimalistic, there are several tools available to perform a more sophisticated microglia morphology quantitation. Is microglia density altered (cells/surface), ramifications, etc. Based on the violin, there seem to be 2 microglia subsets in the OT, one centered around 0.6, the other around 0.8 specificity.

5 In Fig 2, juvenile and adult microglia scRNAseq data are depicted. I assume, its not 100% clear from the text, that the 2 datasets were merged prior to the subclustering depicted in 2E.

6 Pseudotime analysis should be performed to delineate potential developmental trajectories of these microglia subsets in juvenile and adult fish.

7 Overall, the number of microglia in the scRNAseq data set is not very high, 3529. This might preclude a more robust identification of subcluster marker genes. JM0, the largest cluster, did not have a regional enrichment signature. If it is indeed a common microglia subset, as proposed by the authors, it is difficult to imagine why it was not represented in the bulk RNAseq data. In 2C, cells from clusters 0 and 1 are somewhat mixed, were the data possibly over/underclustered or is the # of cells too low for better separation?

Reviewer #3, response: We sincerely appreciate the reviewer's detailed feedback. We first focus on the question regarding the number of sequenced cells and the resolution at which our single-cell data was analyzed, then address each subsequent point chronologically.

The reviewer notes: A weakness is the low number of cells subjected to scRNAseq, somewhat hampering the interpretation of the scRNAseq data, which is inherently less sensitive than bulk RNAseq, which in its turn lacks cellular resolution. For more robust clustering and marker genes etc., more cells would help. **In point #7 the reviewer adds:** Overall, the number of microglia in the scRNAseq data set is not very high, 3529. This might preclude a more robust identification of subcluster marker genes. JM0, the largest cluster, did not have a regional enrichment signature. If it is indeed a common microglia subset, as proposed by the authors, it is difficult to imagine why it was not represented in the bulk RNAseq data. In 2C, cells from clusters 0 and 1 are somewhat mixed, were the data possibly over/underclustered or is the # of cells too low for better separation?

Response: We agree that more cells would have been ideal. The data presented is from all CD45+ cells isolated from 13 fish (10 juvenile, 3 adult). We recovered 9043 total cells that passed our quality control thresholds (6666 juvenile and 2377 adult), of which 3539 juvenile cells and 2080 adult cells were *mpeg1.1+*.

To limit potential microglial activation from prolonged time *ex vivo*, we prioritized speed but recovered fewer cells. However, as we discuss further below, we feel that the conclusions put forth in this manuscript are well supported at this level of resolution, and are further supported by our parallel analysis using bulk-sequencing data.

To address the reviewer's point regarding clustering resolution, we have added new analyses (**Reviewer 3, Fig. 4, from Manuscript Fig. S2H-I**). First, we examine our dataset at multiple clustering resolutions (0.1, 0.3*, and 0.5). Clustering and differential gene expression analysis suggests that there is a meaningful biological difference between clusters 0 and 1 that is statistically significant at this number of sequenced cells. While it is possible to merge clusters 0 and 1 at a low enough resolution (0.1), doing so leads to loss of relevant information. We chose the resolution shown in the

manuscript (0.3), for several reasons. First, at that resolution, differential expression analysis between clusters 0 and 1 suggest a clear separation: 337 genes that were up- or down-regulated by at least 15%, including 50

genes up- or down-regulated by at least 40% (including genes expressed in at least 10% of a cluster; $p < 0.001$; Rev. 3 Fig. 4B). Second, feature plots of DE genes showed a clear gradient in expression between clusters 0 and 1, particularly in genes associated with OT by bulk sequencing (e.g. *bzw2*, *g0s2*; Rev. 3 Fig. 4C). Third, our bulk sequencing eigengene analysis (Manuscript Fig. 4B) indicates a high degree of overlap between OT microglia (which are predominantly in neurogenic regions) and cluster 1, and little overlap with cluster 0. Fourth, RNA velocity analysis with scVelo predicts that cluster 0 may be a precursor state to cluster 1 (Reviewer-only Figure Fig. 4D) (Bergen et al 2020). This strengthens our confidence that a clustering resolution separating clusters 0 and 1 is a better representation of the neurogenic-associated microglial signature.

Notably, the main conclusions of the paper would remain largely unchanged by combining clusters 0 and 1. For example, 625 out of 800 differentially regulated genes between clusters 4 and 0/1 are shared regardless of whether clusters 0 and 1 are pooled or separated. Separating clusters 0 and 1 does not change the overall gene signature observed in neurogenic niche-associated and synaptic-region associated microglia, but rather emphasizes the observed heterogeneity in gene expression within the non-synaptic region associated microglial population.

The reviewer correctly points out that we were unable to identify the source of microglial cluster 0. There are multiple possible explanations for this. Cluster 0 may represent a microglial state found throughout the brain, and therefore not enriched in markers for any one brain region found in our bulk sequencing analysis. It is also possible that cluster 0 microglia are found in a brain region not included in our bulk sequencing experiment. We have noted this in our results and discussion.

- Bergen, V., Lange, M., Peidli, S. *et al.* Generalizing RNA velocity to transient cell states through dynamical modeling. *Nat Biotechnol* **38**, 1408–1414 (2020). <https://doi.org/10.1038/s41587-020-0591-3>

1. Last year, a paper reporting on microglia heterogeneity in the human fetal brain was published in *Science*, using scRNAseq. The authors should determine the (dis)similarity of the NAMs and SAMS to the (phagocytic) microglia clusters in the human fetal CNS to compare microglia development in the zebrafish and human brains are.

Response: We have now performed a comparison between the gene signatures observed in the Kracht et al 2020 human fetal microglial analysis and our juvenile microglial dataset (Rev 3, Fig. 5; Manuscript figure S6B). We loaded the differentially expressed human gene list (Kracht et al. 2020, Table S3) and converted human gene IDs to their zebrafish homologs using biomaRt. We found that 30% (304/1054) of the human genes were not annotated to zebrafish homologs and were therefore excluded from the analysis. We then calculated an eigengene value for each

human fetal cluster in the zebrafish juvenile clustering and used a non-parametric analysis of variance to look for evidence of significant enrichment of human fetal cluster signatures within the zebrafish dataset. The

results are represented in a heatmap showing the estimated enrichment in each human fetal signature in a given zebrafish cluster compared to all other zebrafish clusters (number/color = estimated difference in eigengene expression; * $p < 10^{-50}$, ** = $p < 10^{-100}$, *** = $p < 10^{-200}$). We found a strong correlation between human fetal cluster 6 (dividing cells) and zebrafish cluster JM2 (dividing cells); additionally, human fetal clusters 3 (GW11-12-enriched) and 5 (immediate early gene-expressing microglia) were found to share some similarities with NAM cluster JM1, while the smaller and relatively un-annotated human fetal cluster 11 (MRPL23+ microglia) was associated with SAM cluster JM4. From this we conclude that NAM-like cells can be found in human fetal microglia (>20% of cells are found in clusters HF3 and HF5). However SAMs, if present, are rare (<3%). This could suggest that SAMs are not abundant at the developmental stages represented in that dataset, or alternately, that in the human brain, these functions do not segregate to a specific subcluster. This is possible, as cell bodies and synapses are intermingled in most of the mammalian brain but spatially segregated in the zebrafish. Nevertheless, the synapse-associated genes identified in this manuscript are expressed in human brain and may serve similar synaptic functions in mammals.

- Kracht, L., Borggrewe, M., Eskandar, S., Brouwer, N., Chuva de Sousa Lopes, S.M., Laman, J.D., Scherjon, S.A., Prins, J.R., Kooistra, S.M., and Eggen, B.J.L. (2020). Human fetal microglia acquire homeostatic immune-sensing properties early in development. *Science* (80-.). 369, 530–537.

2. In fig 1A, the DAPI signal is oversaturated, and individual nuclei cannot be distinguished. This makes it a bit difficult to interpret the & of mpeg-expressing microglia/total cell number.

Response: Thank you for this comment. We have updated these images to better resolve the DAPI signal.

3. In fig 1G, 2 “typical” microglia are depicted, more convincing would be, to depict the microglia located at the center/middle of the violins depicted in 1H. In 1C, fairly ramified cells can be seen in the OT.

Response: Thank you for this comment. We have clarified this figure to better illustrate our point. We typically observe a somewhat bimodal distribution, wherein the relatively rare microglia found in OT synaptic regions are just as ramified as HB microglia, but the majority of OT microglia are clustered in neurogenic regions and ameboid. For this reason, we did not feel that showing a cell at the exact midpoint best represented our observations. We have clarified the image as follows: 1) we added the exact sphericity index calculated for the representative images shown, and 2) we now clarify that the image on the left represents an ameboid cell found in the OT, whereas the right shows a ramified cell found in the hindbrain, to avoid misrepresenting these as a ‘typical’ or average result.

observe a somewhat bimodal distribution, wherein the relatively rare microglia found in OT synaptic regions are just as ramified as HB microglia, but the majority of OT microglia are clustered in neurogenic regions and ameboid. For this reason, we did not feel that showing a cell at the exact midpoint best represented our observations. We have clarified the image as follows: 1) we added the exact sphericity index calculated for the representative images shown, and 2) we now clarify that the image on the left represents an ameboid cell found in the OT, whereas the right shows a ramified cell found in the hindbrain, to avoid misrepresenting these as a ‘typical’ or average result.

4. The morphometrics depicted in 1H is a bit minimalistic, there are several tools available to perform a more sophisticated microglia morphology quantitation. Is microglia density altered (cells/surface), ramifications, etc.

Response: To address the reviewer’s point, we now add new data showing the results of Sholl analysis based on maximal projection images, an independent morphologic approach (Rev 3, Fig. 6, Manuscript Fig. 11). These results are consistent with morphological findings in figure 1H and provide additional support that HB microglia are more ramified. We also show that microglia differ in distribution and location, where in the OT microglia predominately cluster around neurogenic zones and in the HB synaptic regions.

5 In Fig 2, juvenile and adult microglia scRNAseq data are depicted. I assume, its not 100% clear from the text, that the 2 datasets were merged prior to the subclustering depicted in 2E.

Response: Thank you for the comment. Yes, they were merged before subclustering using the Harmony package in R. We have now clarified this in the methods, figure legend, and text.

6. Pseudotime analysis should be performed to delineate potential developmental trajectories of these microglia subsets in juvenile and adult fish.

Response: Thank you for this comment. We have now performed pseudotime analysis (**Rev. 3, Fig 7. Manuscript Fig. S6A**) calculated using Monocle 3 (Cao et al 2019). This predicts that both NAMs (JM1) and SAMS (JM4) are endpoints, later in pseudotime than cluster JM0 and dividing cells. Additionally, adult-specific cluster A4 is later in pseudotime than other clusters, corresponding to its emergence after the juvenile timepoint. Diffusion mapping with Destiny (Angerer et al 2016) also suggests that NAMs and SAMs represent endpoints rather than intermediate stages of microglial development (diffusion coordinates 3 and 4 are shown as they provide the best separation between microglial-specific clusters). Both analyses suggest that NAMs and SAMs are endpoints and potentially terminally differentiated microglial subsets derived from a pool of microglia composed of both cycling and non-cycling cells.

- Philipp Angerer, Laleh Haghverdi, Maren Büttner, Fabian J. Theis, Carsten Marr, Florian Buettner, *destiny*: diffusion maps for large-scale single-cell data in R, *Bioinformatics*, Volume 32, Issue 8, 15 April 2016, Pages 1241–1243, <https://doi.org/10.1093/bioinformatics/btv715>
- Cao, J. et. al. The single-cell transcriptional landscape of mammalian organogenesis. *Nature* 566, 496-502 (2019). <https://doi.org/10.1038/s41586-019-0969-x>

Reviewers' Comments:

Reviewer #1:

Remarks to the Author:

This is a lovely and important study. I particularly appreciate the thoughtful responses to the first round of reviews. I have no further comments to make on this manuscript and I look forward to its publication. Congratulations to the authors for their fine work.

Bruce Appel

Reviewer #3:

Remarks to the Author:

In the revised manuscript by Silva and co-workers, additional experiments and clarifications were added. Almost all my suggestions were addressed, and additional analyses incorporated. My comments regarding the number of cells analysed by snRNAseq is satisfactorily addressed and the conclusions presented are supported by the data.

In short, an elegant study, important for the field and I endorse acceptance.